# Transcriptomic Analysis Reveals the Temporal and Spatial Changes in Physiological Process and Gene Expression in Common Buckwheat (*Fagopyrum esculentum* Moench) Grown under Drought Stress

**Zehao Hou [1,†], Junliang Yin [1,†], Yifei Lu [1], Jinghan Song [1], Shuping Wang [1], Shudong Wei [2], Zhixiong Liu [3], Yingxin Zhang [1] and Zhengwu Fang [1,\*]**

[1] College of Agriculture, Yangtze University, Jingzhou 434000, Hubei, China; 201771374@yangtzeu.edu.cn (Z.H.); yinjunliang@yangtzeu.edu.cn (J.Y.); 201872410@yangtzeu.edu.cn (Y.L.); 201875095@yangtzeu.edu.cn (J.S.); wangshuping@yangtzeu.edu.cn (S.W.); 518009@yangtzeu.edu.cn (Y.Z.)

[2] College of Life Science, Yangtze University, Jingzhou 434000, Hubei, China; 500896@yangtzeu.edu.cn

[3] College of Horticulture and Gardening, Yangtze University, Jingzhou 434000, Hubei, China; zxliu77@yahoo.com

\* Correspondence: fangzhengwu88@yangtzeu.edu.cn; Tel.: +86-137-9732-9339

† These authors contributed equally to this work.

**Abstract:** Common buckwheat is a traditional alternative crop that originated from the northwest of China and is widely cultivated worldwide. However, common buckwheat is highly sensitive to drought stress, especially at the seedling stage, and the molecular mechanisms underlying the response to drought stress still remain elusive. In this study, we analyzed the stress phenotypes of buckwheat seedlings under drought condition. The results showed the wrinkled cotyledon due to the decrease of relative water content (RWC) in response to the increased activity of antioxidant enzymes. Transcriptomic analysis was further performed to analyze the regulation patterns of stress-responding genes in common buckwheat cotyledons and roots under drought stress conditions. Characterizations of the differentially expressed genes (DEGs) revealed differential regulation of genes involved in the photosynthesis and oxidoreductase activity in cotyledon, and that they were highly related to the post-transcriptional modification and metabolic process in root. There were 180 drought-inducible transcription factors identified in both cotyledons and roots of the common buckwheat. Our analysis not only identified the drought responsive DEGs and indicated their possible roles in stress adaption, but also primarily studied the molecular mechanisms regulating the drought stress response in common buckwheat.

**Keywords:** common buckwheat; cotyledon; root; drought stress; transcriptome analysis

## 1. Introduction

Among the forms of environmental stress, drought stress has been considered as one of the major constraints in plant growth, survival, and production [1,2]. A lack of water not only disturbs photosynthesis, limits metabolic reactions, and inhibits $CO_2$ exchange, but also results in stress-related damage to chloroplasts [3–5]. In order to adapt to the extreme environments, plants have evolved several mechanisms (e.g., drought escape, avoidance, and tolerance) to ensure high survival rates under drought stress [5,6]. Specifically, plants recruit a variety of responding mechanisms to deal with drought stress [7–9], such as stomatal closure, leaf rolling, and alteration in biosynthetic and antioxidant pathways, which are highly regulated by complex transcriptional networks [10].

Drought stress affects several physiological and biochemical pathways in plants [11]. Previous research has shown that the water deficits not only affect the chlorophyll biosynthesis, but also the level of malondialdehyde (MDA) and the relative water contents (RWC) of the plant, brining detrimental effects to the lipid peroxidation, and membrane constitution [12,13]. Also, the abiotic stresses can further induce the oxidative stress through generating reactive oxygen species (ROS), a prevalently recognized destroyer in cellular metabolism [14–16]. ROS generate the oxidation of photosynthetic pigments, initiate lipid peroxidation, and degrade proteins in plants, and thereby cause damage to cell structures and metabolism, particularly those associated with photosynthesis [17,18]. To counteract the effects of oxidative stress, plants have developed an efficient detoxification defense system consisting of non-enzymatic scavengers and enzymatic components to scavenge free ROS [19,20]. In terms of the enzymatic scavenging, a series of antioxidative enzymes, including peroxidase (POD), catalase (CAT), superoxide dismutase (SOD), and ascorbate peroxidase (APX), have been reported to play a vital role in reducing the damage effects (i.e., water deficiency) caused by drought stress [21]. There is evidence that keeping a high antioxidative enzyme activity level to reduce the damaging effects caused by water deficit stress may be associated with the osmotic stress tolerance of plants [22], which is also found to be positively related to plant drought tolerance [23].

Presently, there are several drought-inducible genes, including stress responses and resistance, which have been identified through transcriptome analyses [1]. These genes can be divided into two groups according their functions. The first group is composed of function proteins that include the late embryogenesis abundant (LEA) proteins, ROS detoxification enzymes, molecular chaperones, heat shock proteins (HSP), and lipid-transfer proteins [24,25]. The second group is involved in regulatory proteins or transcriptional factors (TFs), which correlate with the signal transduction and stress-responsive gene expressions, for example, the phospholipases and dehydration-responsive elements [26,27]. In order to elucidate the biological functions of these genes, several transgenic plants overexpressing various drought-resistant genes have been generated, which have both shown enhanced drought tolerance and growth retardation [28–31], demonstrating that plants may adapt to the drought environment at the expense of normal growth [1].

Common buckwheat (*Fagopyrum esculentum* Moench) is an important dual-purpose alternativecrops originated from Yunnan Province of China [32] and is widely cultivated around the world, especially in China, Japan, and Russia [33]. Because of its abundant nutrients in seeds, common buckwheat is considered as one of the sources of flour, groats, and whole grain foods. However, common buckwheat is highly sensitive to drought, especially at the seedling stage [34,35], and short-term drought occurs frequently in China, posing a threat to domestic food safety [36]. Thus, it would be important and necessary to study the physiological and molecular bases of osmotic stress tolerance in common buckwheat. We have previously identified *FeDREB1L* (GenBank: JN600617.1), a CBF/DREB homologous gene, from common buckwheat, and overexpression of the *FeDREB1L* gene was found to significantly increase the water deficit resistance of transgenic *Arabidopsis* [31]. In order to further understand the drought-resistant mechanism and identify novel water-deficit-related genes in common buckwheat, a transcriptomics analysis was carried out to investigate the variations in common buckwheat growth under short-term drought treatment, and the phenotypes and biochemical traits of seedlings were also analyzed. Our results may provide more information with regard to the transcriptional control of common buckwheat under the abiotic stresses, and help to identify the novel genes that are potentially valuable for future common buckwheat breeding.

## 2. Materials and Methods

### 2.1. Plant Material and Drought Treatments

Common buckwheats (cv. Xi'nong 9976) were germinated in Petri dishes in an incubator (plant growth incubator JY412L, Shanghai, China) in darkness (25 °C) and relative humidity of approximately 60%. After germination for 36 h, when the root length of the seedlings grew to approximately 2 cm,

the seedlings were transplanted for hydroponics in an incubator with 12 h photoperiods (25 °C/20 °C, day/night temperature) and relative humidity of approximately 60%. The 7-day old buckwheat seedlings were treated with 15% polyethylene glycol 6000 (PEG 6000) solution for 1 d, 3 d, and 5 d. After treatment, the cotyledons and roots were collected and quickly frozen in liquid nitrogen and stored at −80 °C until used. The seedlings before drought treatment were served as the control. Each treatment was carried out in three biological replicates.

*2.2. Physiological Measurement*

Relative water content (RWC) was determined according to the formula described by Pan et al. [36]. The chlorophyll content and chlorophyll a/b ratio was calculated using to the method described by Harper et al. [37]. The changes of malondialdehyde (MDA) concentration were determined using the thiobarbituric acid (TBA) reaction [38], and the activities of POD and CAT were detected according to the description of Harper et al. [37]. The Rubisco activities were assayed with Rubisco assay kits (Beijing Solarbio Science and Technology Co., Ltd., Beijing, China) according to the manufacturer's instructions.

*2.3. RNA Isolation and Transcriptome Sequencing*

Total RNA was isolated from the non-treated control and drought-stressed cotyledon and root samples using EASYspin Plus Plant RNA Kit (Aidlab, Wuhan, China). The RNA quality was checked by Agilent bioanalyzer 2100 (Agilent Technologies, Santa Clara, CA, USA), and Nanodrop 2000 r spectrophotometer (Nanodrop Technologies, Wilmington, DE, USA) was used for RNA quantification.

Transcriptome sequencing was performed at Beijing Allwegene Technology Co. Ltd. (Beijing, China), following manufacturer protocols. Briefly, mRNA was enriched from total RNA using Oligo (dT) magnetic beads, and the mRNA was fragmented into small pieces using a fragmentation buffer. Then, these fragments were used as reverse transcription to synthesized the first- and second-strand cDNA, and the second-stand cDNA were purified with AMPure XP Beads Kit, repaired, poly (A) added, and ligated to paired-end adapters. Finally, the cDNA libraries were sequenced on Illumina HiSeqTM 2500 platform. Each sample had three biological replicates. The raw reads were submitted to the National Center for Biotechnology Information (NCBI) Sequence Read Archive with a Bioproject ID: PRJNA555746.

The raw reads in FASTQ format were processed using in-house Perl scripts, and the high-quality clean data were obtained by removing the low-quality data, which included the reads that contained the adapter, and more than 10% of N nucleotides, and the low-quality reads that contained more than 50% of low quality bases (Q-value ≤ 20). In addition, we calculated the Q30, GC content, and sequence duplication levels for the clean data. Cleaned and qualified reads were aligned against the *F. esculentum* reference genome [39] using Tophat2 software [40]. Then, these sequences were subjected to functional annotation and coding sequence (CDS) prediction [41], and the resulting sequences were called genes. Finally, fragments per kilobase of transcript permillionmapped reads (FPKM) method was used to calculate the gene expression unit.

*2.4. Identification and Functional Annotation of Different Expressed Genes (DEGs)*

The differential gene expression analysis was carried out using DESeq software, and DEGs were determined by combining a *q* value cutoff of 0.05 and adjusting to |log2 (fold change)| ≥ 1. For DEG functional annotation, Gene Ontology (GO) enrichment analysis was carried out by GOseq software, and Kyoto Encyclopedia of Genes and Genomes (KEGG) was used to perform pathway enrichment analysis of DEGs. In addition, the gene expression profiles at the pathway were display by MapMan software (version 3.6.0) [33].

*2.5. Quantitative Real-Time PCR (qRT-PCR) Analysis*

Total RNA from cotyledons and roots of both samples were extracted using EASYspin Plus Plant RNA Kit (Aidlab, Wuhan, China) following the manufacturer's protocols, and the first-strand

cDNA for qPCR analysis was synthesized from 500 ng of total RNA using PrimeScript RT Reagent Kit with gDNA Eraser (Takara, Dalian, China) following the manufacturer's instructions, and cDNA was diluted 10-fold and used as the template for qRT-PCR. The primers were designed using Primer Premier 5.0 and beta-actin was used as a reference gene, with the primer information listed in Table S1. qRT-PCR was performed on a CFX 96 real-time PCR system (BioRad, Hercules, CA, USA) using TB Green (TaKaRa), according to the manufacturer's protocols. PCR amplification was conducted in a volume of 20 μL, containing ~100 ng of cDNA template, 0.6 μL of each primer (10 μmol), and 10 μL PCR-mix (2×). The conditions for all reactions were as follows: 30 s at 95 °C, followed by 40 cycles of 10 s at 95 °C, and 30 s at 55 °C, and the melting curve was generated to confirm the PCR specificity. The non-treated control treatment was chosen as the control to standardizing all samples, using the $2^{-\Delta\Delta Ct}$ method to calculate the relative expression levels [42].

### 2.6. Statistical Analysis

A one-way ANOVA was carried out by SPSS Statistics 19.0 software (IBM Corp, Armonk, NY, USA), and means were compared using the Duncan test to determine significant differences ($p < 0.05$). The results were presented as mean ± SD (standard deviation).

## 3. Results

### 3.1. Changes in Phenotype of Common Buckwheat Seedlings at Drought Stress

To investigate the dynamic phenotypic changes of common buckwheat seedlings in response to drought stress treatments, plant height, root length, and relative water content (RWC) were measured under 15% PEG 6000 solution treatments across four time-points (0, 1, 3, and 5 days). There was no significant change in plant height, root length, and RWC of control samples after 1, 3, and 5 days (Table S2). Under drought stress, the buckwheat seedlings showed stress phenotypes of wrinkled cotyledon (Figure 1a), but the plant height and root length did not significantly change during the treatment (Figure 1b,c). The RCW is generally used as an important indicator of plant water status under osmotic conditions, and in this study, the RCW values were clearly decreased in the 3 and 5 day-treated (DPT3d and DPT5d) seedlings, but were not significantly different among the control (CK) and the 1 day-treated (DPT1d) seedlings (Figure 1d).

### 3.2. Changes in Physiology of Common Buckwheat Seedlings under Drought Conditions

To investigate the physiological changes under different levels of water deficit conditions, the MDA content and the activities of POD and CAT of cotyledons were measured after drought treatment. Compared with the drought treatment, these physiological traits were not significantly changed during the whole treatment under control conditions, in which the content of chlorophyll a and chlorophyll b were increased in the 1, 3, and 5 day control plants; however, the chlorophyll a/b ratios were not significantly changed during the whole treatment under the control condition (Table S3). Under water deficit condition, the MDA content was greatly increased from 0 to 1 days, and then slightly increased until 5 days (Figure 2a). Meanwhile, the activities of POD and CAT were significantly increased under PEG treatment (Figure 2b,c). The chlorophyll a content was elevated in the 3 and 5 dy-treated plants (Figure 2d), while the content of chlorophyll b content was increased in the 1, 3, and 5 day-treated plants (Figure 2e). In addition, there was no difference between the control and 1 and 3 day-treated seedlings in chlorophyll a/b ratios, but marked decreases were observed in the 5 day-treated plants (Figure 2f). These results indicated that there were significant changes in the physiology of the common buckwheat seedlings in response to osmotic stress.

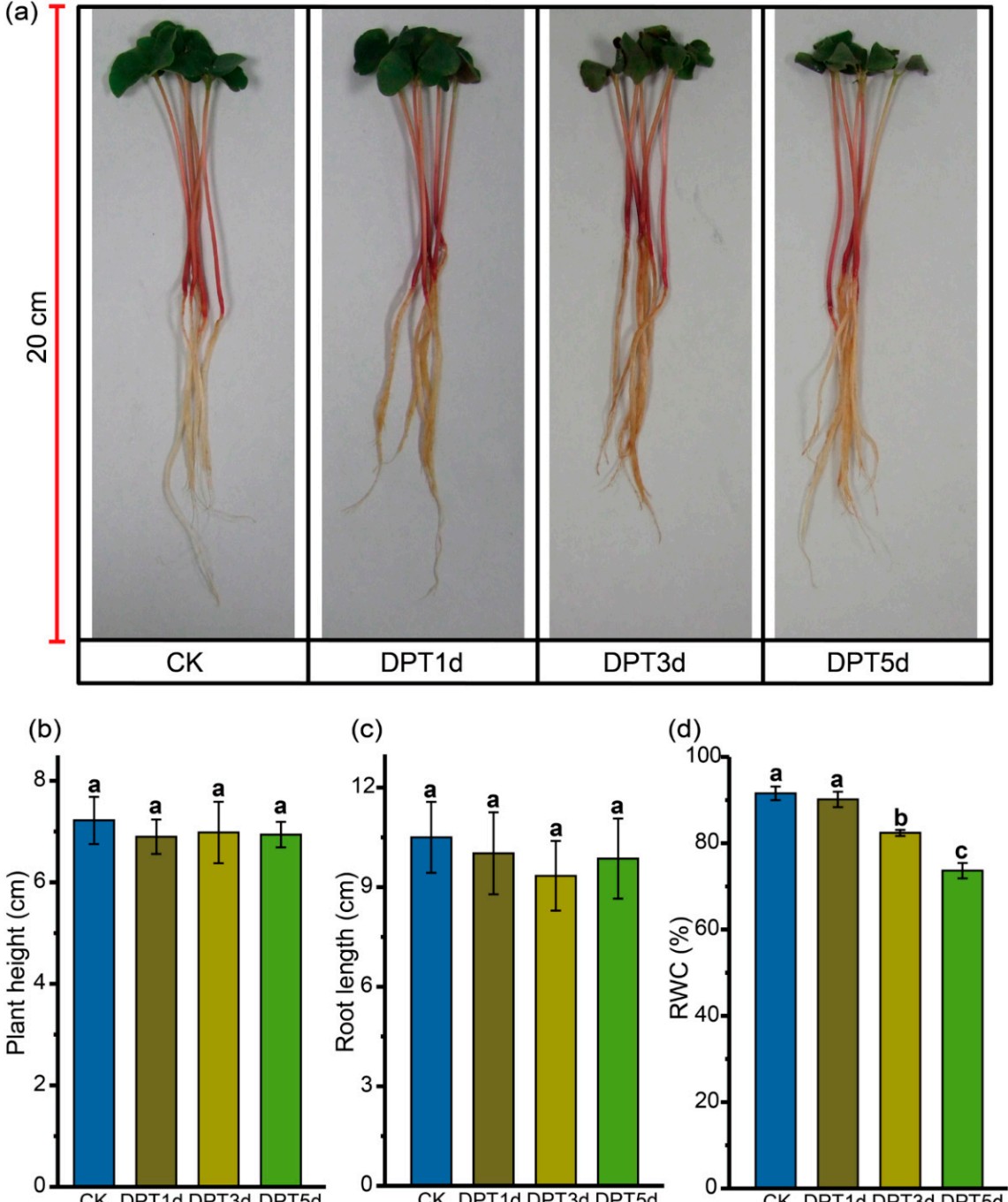

**Figure 1.** Changes in phenotype of common buckwheat seedlings under drought stress. (**a**) A photograph of common buckwheat seedlings after PEG treatments; CK, non-stressed control; DPT1d, DPT3d, and DPT5d, drought treatment with PEG solution for 1, 3, and 5 days, respectively; bar = 20 cm. The change of (**b**) plant height, (**c**) root length, and (**d**) relative water content of cotyledon during drought stress treatment. Bars represent means of three replicates ± SD (standard deviation). Different letters indicate means that are significantly different at the $p < 0.05$ level among different drought conditions.

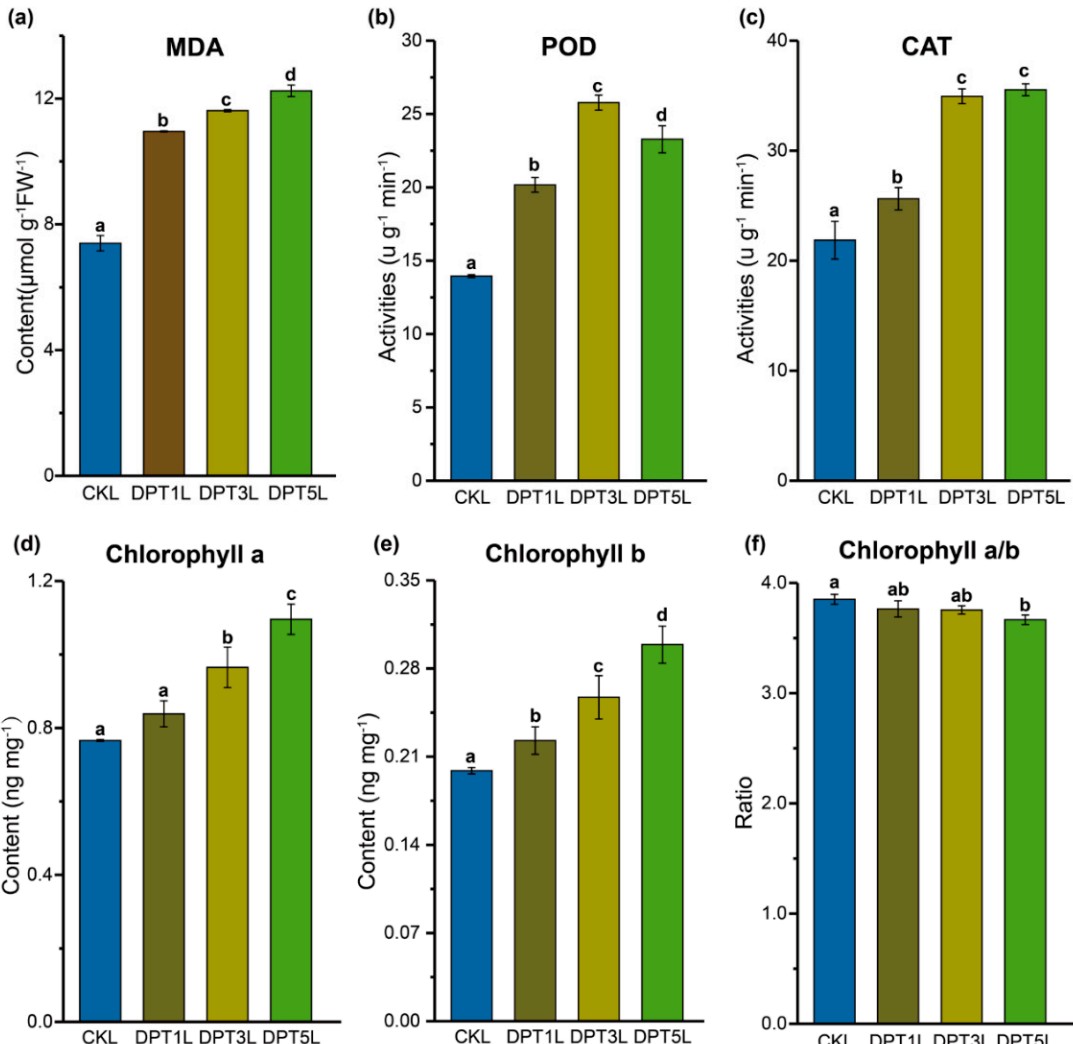

**Figure 2.** Changes in physiology in cotyledons of common buckwheat seedlings under different drought stress conditions. (**a**) Changes in the malondialdehyde (MDA) content of cotyledons, (**b**) changes in the peroxidase (POD) activities of cotyledons, (**c**) changes in the catalase (CAT) activities of cotyledons, (**d**) chlorophyll a content in the cotyledons, (**e**) chlorophyll b content in the cotyledons, and (**f**) ratios of chlorophyll a/b.

### 3.3. Overview of the Common Buckwheat Transcriptome and Identification of DEGs

To reveal the expression changes in common buckwheat cotyledons and roots at 0, 1, 3, and 5 days after PEG treatments, 24 common buckwheat samples (including 12 cotyledon samples and 12 root samples) were used for RNA-Seq analysis to further investigate the changes at the transcriptional level. A full-scale sequencing analysis from 24 cDNA samples is shown in Table S4, 88.76 gigabytes (Gb) of clean reads were abstained, and the percentages of the Q30 base of these 24 common buckwheat samples were greater than or equal to 95.70%. Furthermore, there was a highly mapped efficiency between the samples and reference genome (79.46%–87.91%), which met the requirements for information analysis. There were 877,111 CDS (coding sequence)-encoded proteins, and the length of CDS is shown in Figure 3a. Of these CDSs, only a minority (3729 CDSs, 0.43%) were more than 1500 nt, and 90.57% of CDSs appeared with a length ranging from 0 to 500 nt. In addition, the principal component analysis (PCA) was performed and the results demonstrated that the control treatment was clearly separated from the drought stress treatment in the cotyledons or roots (Figure 3b), suggesting that the gene expression pattern of common buckwheat was greatly changed under drought condition.

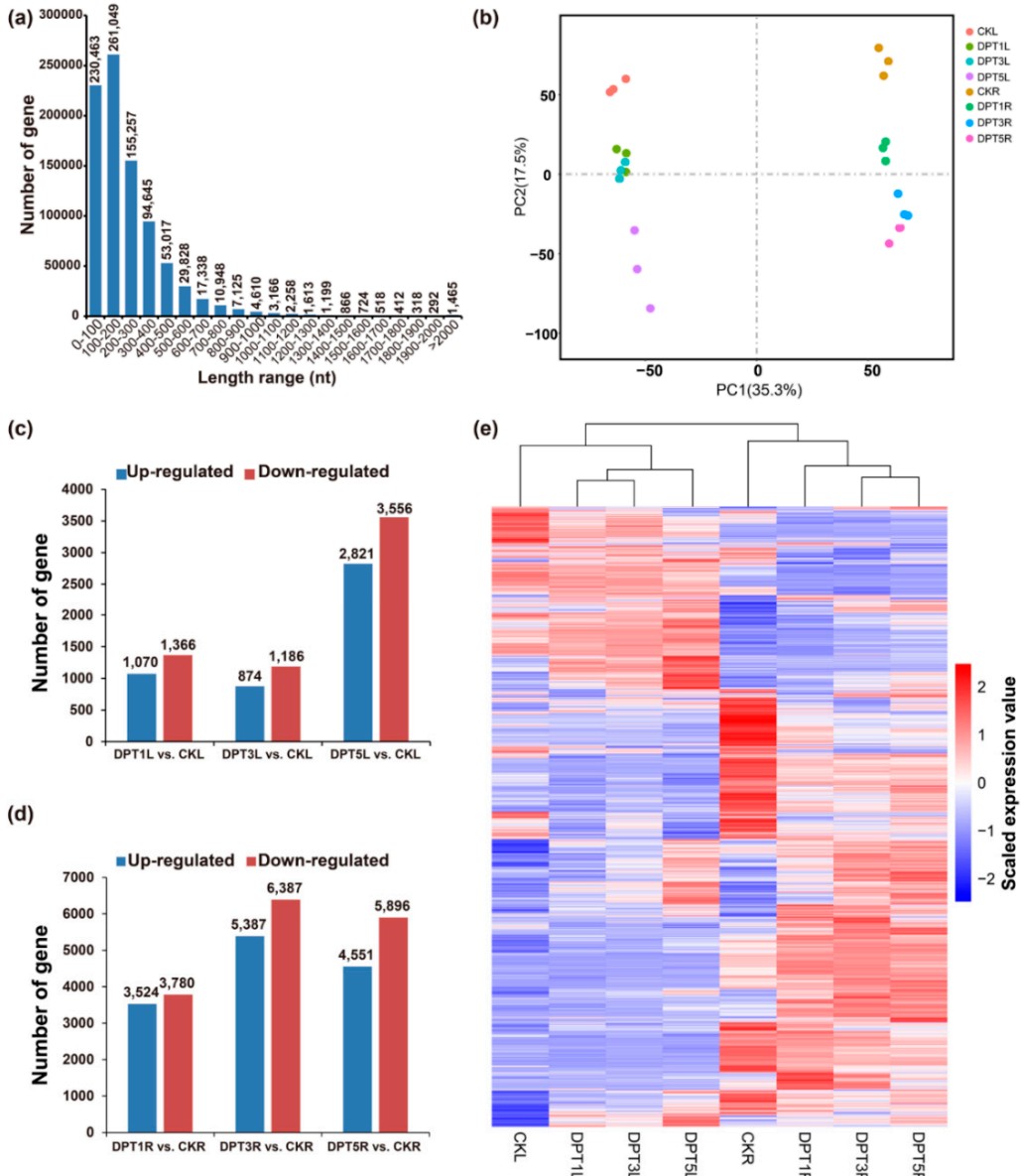

**Figure 3.** Overview of the transcriptomic results and changes in gene expression profiles in cotyledons and roots after drought stress treatment. (**a**) Predicted length distribution map of coding sequence (CDS)-encoded protein nucleotide (nt). (**b**) Principal component (PC) analysis of gene expression at different drought stress conditions. (**c**) Numbers of differently expressed genes (DEGs) in cotyledons of common buckwheat seedlings at different drought stress conditions in pairwise comparisons. (**d**) DEGs in roots of common buckwheat seedlings at different drought stress conditions in pairwise comparisons. (**e**) Heat-map graphics exhibiting the gene expression levels of total DEGs. CKL and CKR are the cotyledon samples and root samples of the non-stressed control, respectively. DPT1L, DPT3L, and DPT5L are the cotyledon samples of drought treatment for 1, 3, and 5 days, respectively. DPT1R, DPT3R, and DPT5R are the root samples of drought treatment for 1, 3, and 5 days, respectively. DPT1L vs. CKL, DPT3L vs. CKL, and DPT5L vs. CKL are the cotyledon samples of drought treatment for 1, 3, and 5 days compared to the non-stressed control, respectively. DPT1R vs. CKR, DPT3R vs. CKR, and DPT5R vs. CKR are the root samples of drought treatment for 1 day compared to the non-stressed control, respectively. Up-regulated means that genes were up-regulated in drought stress conditions compared to the non-stressed control and down-regulated means that genes were down-regulated in the drought stress condition compared to the non-stressed control.

The gene expression levels were calculated as FPKM values via HTSeq software analysis, and the differential gene expression analysis was carried out using DESeq software. There were 2436, 2060, and 6377 DEGs identified in the cotyledons after drought treated for 1, 3, and 5 days, respectively (Figure 3c). In the roots, compared with the control treatment, 7304, 11,774, and 10,447 DEGs were identified after drought treatment for 1, 3, and 5 days, respectively (Figure 3d). Upon drought stress exposure, more DEGs were identified in the roots than in cotyledons, suggesting that there were different drought stress response mechanisms between roots and leaves in common buckwheat. Furthermore, in order to provide a comprehensive understanding of the change in gene expression of common buckwheat under drought conditions, a heat map was developed, as shown in Figure 3e, to exhibit the overall changes of the gene expression under water deficit conditions.

*3.4. Comprehensive Sets of DEGs in the Cotyledons of Drought-Treated Common Buckwheat Seedlings*

Venn diagrams showed that the number of genes commonly up-regulated under the DPT1L and DPT5L were greater than the number of genes commonly up-regulated under the DPT3L and DPT5L, and the number of commonly down-regulated genes showed the same trend (Figure 4a,b). Gene Ontology (GO) was used to find the functional significance of the identified DEGs (Figure 4c), and the GO terms related to signaling and DNA modification were detected in the set of genes up-regulated under the drought conditions, while the GO terms related to light harvesting and light reaction were detected in the sets of genes down-regulated under the water deficit conditions. Furthermore, the DEGs related to the light reactions of photosynthesis and the Calvin cycle were visualized through MapMan analysis (Figure 4d), and the expression of most of these DEGs was decreased in the DPT3L and DPT5L, compared with control treatment. RubisCO, as the major photosynthetic enzyme in plants, plays a crucial role in photosynthesis of green plants. In this study, the activities of RubisCO were significantly decreased at 3 days and greatly declined at 5 days under drought treatment (Figure 4e). These results indicate that photosynthesis in the cotyledons of the common buckwheat seedlings decreased under drought stress conditions. The representative genes related to ABA (abscisic acid) metabolism are listed in Table S5 according to their functional description, and most of these (including 6 *NCED* (*9-cis-epoxycarotenoid dioxygenase*), 3 B3 domain-containing protein, and 3 protein phosphatase 2C) were significantly up-regulated in DPT5L, which indicated that common buckwheat seedlings may use the ABA regulatory systems to affect leaf wilting and defense against the water-deficit stress.

To confirm and investigate the transcriptomic data, qRT-PCR was performed to check the expression levels of several genes, and the expression of *LCHb*, a gene encode chlorophyll a/b binding protein, was markedly decreased under DPT3L and DPT5L conditions (Figure 4f). The expression of *NCED*, a gene involved in ABA biosynthesis, was observably increased under drought stress conditions (Figure 4i). Furthermore, the *DREB1L* gene, which encodes a stress tolerance-related protein, was markedly enhanced in its expression level under drought stress treatment, compared with the non-stressed control plants (Figure 4m). The correlation coefficient ($R^2$) between RNA-Seq data and qPCR results for the 24 total plots was 0.8743 (Figure S1). These analyses of gene expression confirmed that the transcriptomic datasets were efficacious (Figure 4f–n).

*3.5. Comprehensive Sets of DEGs in the Roots of Drought Treated Common Buckwheat Seedlings*

Three comparison groups were constructed to further understanding the universal response in root of common buckwheat to drought stress. As shown in Figure 5a, 1731 genes were both up-regulated in DPT1R, DPT3R, and DPT5R, compared with CKR. Furthermore, 2857 genes were collectively down-regulated in DPT1R, DPT3R, and DPT5R (Figure 5b). All non-overlapped DEGs in the three comparison groups were subjected to GO enrichment analysis, and 430, 606, and 621 GO accessions classified into three categories comprising "molecular function", "biological process", and "cellular component" were identified in DPT1R vs. CKR, DPT3R vs. CKR, and DPT5R vs. CKR, respectively (Table S6). The drought-induced DEGs were mainly involved in the nucleotide binding,

ATP binding, macromolecule modification, protein phosphorylation, protein modification, protein metabolic process, and cellular protein metabolic process (Figure 5c). According to Mapman software analysis, there were 92 protein modification and phosphorylation-related DEGs that were up-regulated after drought treatment (Table S7), suggesting that they were candidate genes for protein modification in roots of drought-treated seedlings.

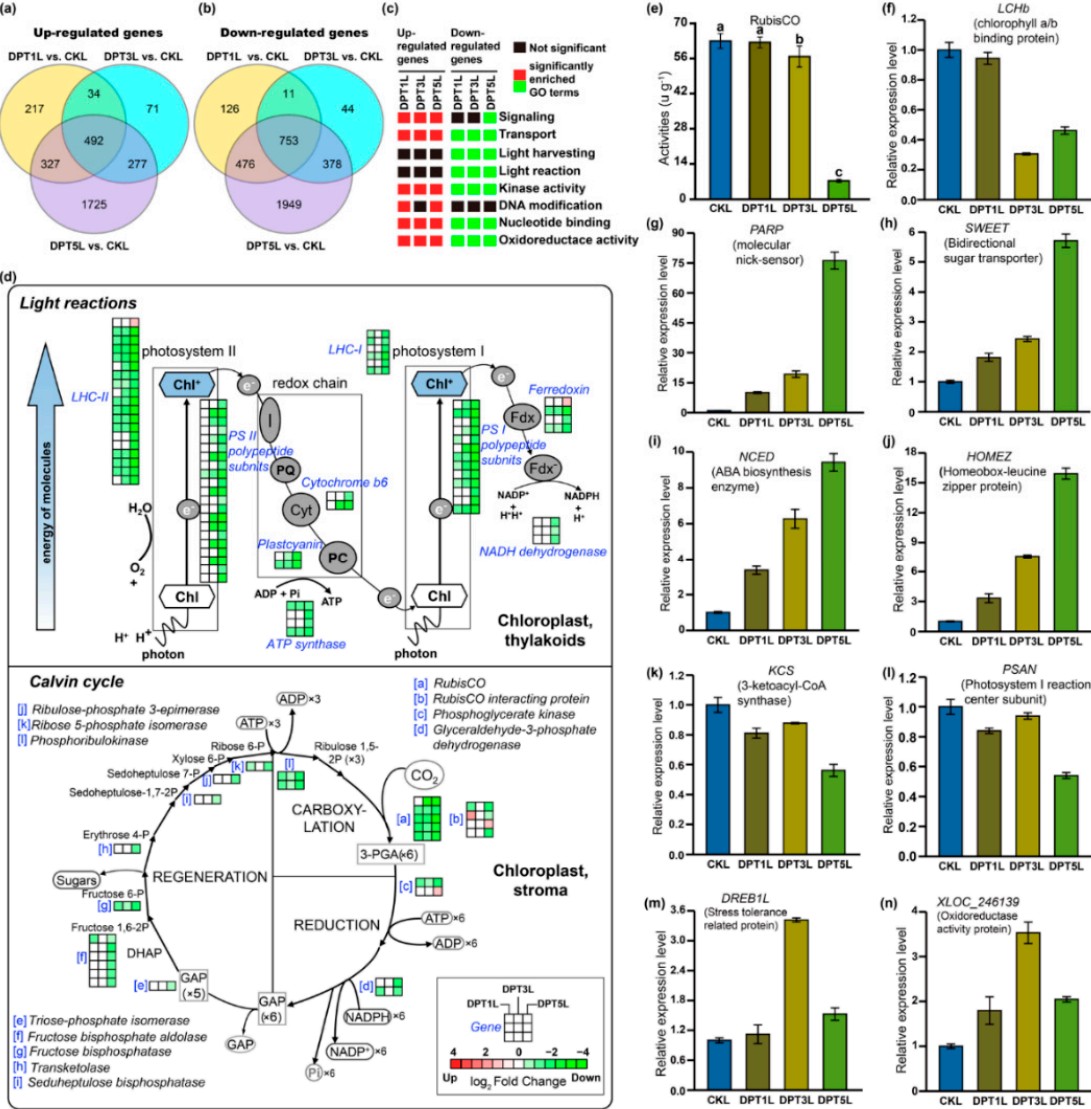

**Figure 4.** Comprehensive expression patterns of DEGs in the cotyledons of common buckwheat seedlings under drought stress conditions. (**a**) Venn diagrams of the numbers of up-regulated [|log2 (Fold Change)| > 1 and q-value < 0.005] genes acquired through the transcriptome analysis. (**b**) Venn diagrams of the numbers of down-regulated genes acquired through the transcriptome analysis. (**c**) Over-represented Gene Ontology (GO) terms estimated using GOseq software. (**d**) Changes in the expression of photosynthesis-related genes. Pathway diagram of light and dark reactions of photosynthesis with superimposed color-coded squares showing DEGs, drawn using MapMan. (**e**) Changes in the RubisCO activities of cotyledons. (**f–n**) Expression profiles of the selected DEGs, *LCHb* (**f**), *PARP* (**g**), *SWEET* (**h**), *NCED* (**i**), *HOMEZ* (**j**), *KCS* (**k**), *PSAN* (**l**), *DREB1L* (**m**), and *XLOC_246139* (**n**) determined using qRT-PCR analyses.

The expression patterns of several genes related to stress tolerance in the roots were analyzed via qRT-PCR (Figure 5d–k). The expression of *AAO*, a gene encoding L-ascorbate oxidase that plays a crucial role in plant cell growth, was markedly decreased under drought stress conditions (Figure 5h),

and the expression of some transcription factors, such as *HOMEZ* and *DREB1L*, were significantly induced by water deficit (Figure 5e,j). In addition, the $R^2$ between the two experiments was 0.9385 (Figure S2). These results confirmed the effectiveness of the transcriptomic datasets and indicated that drought stress strongly affected the expression level of the genes that are related to stress tolerance in the roots.

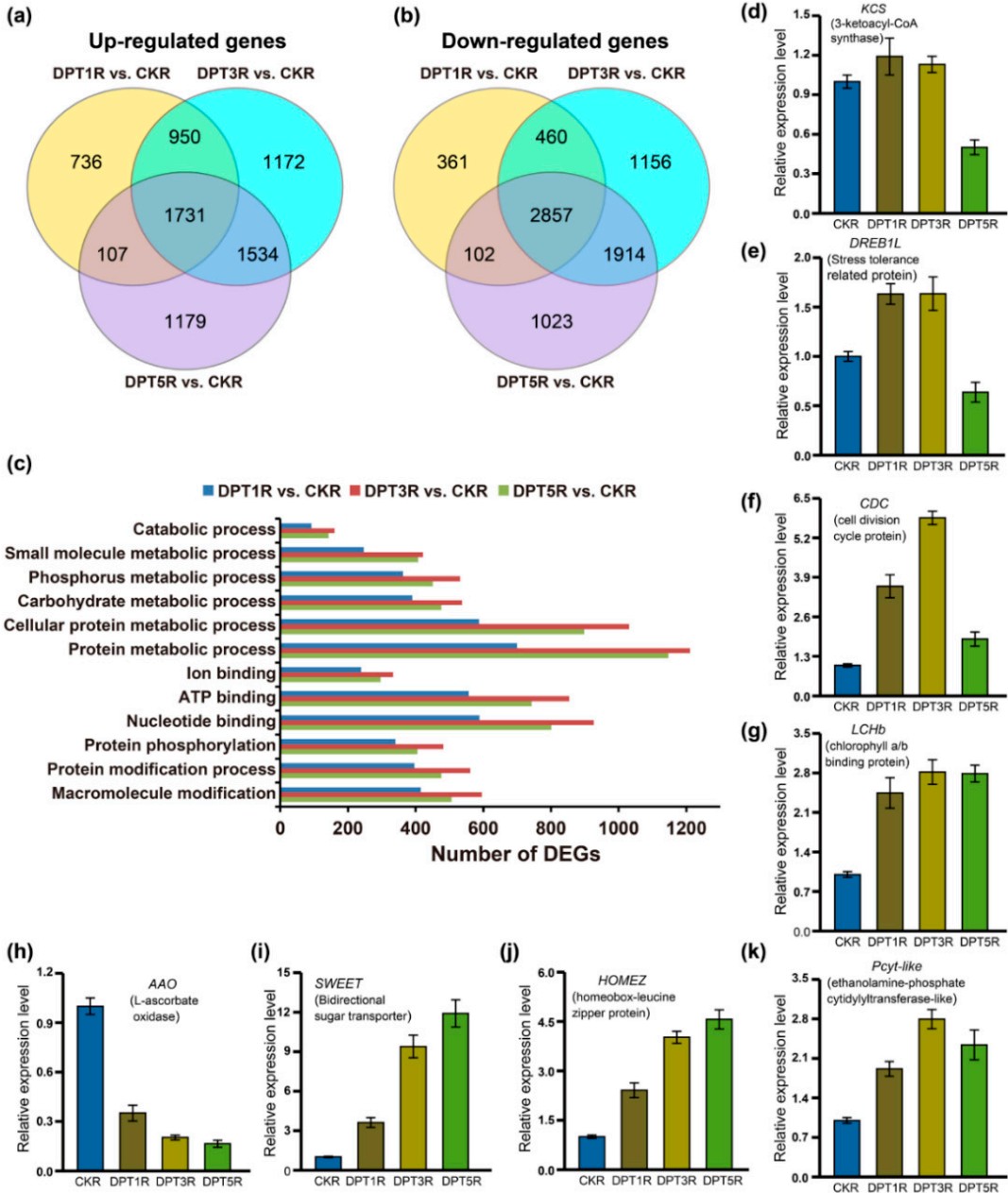

**Figure 5.** Comprehensive expression patterns of DEGs in the roots of common buckwheat seedlings under drought stress conditions. (**a**) Venn diagrams of the numbers of up-regulated [|log2 (Fold Change)| > 1 and qvalue < 0.005] genes acquired through the transcriptome analysis. (**b**) Venn diagrams of the numbers of down-regulated genes acquired through the transcriptome analysis. (**c**) Classification of DEGs based on metabolism, binding and modification categories. (**d–k**) Expression profiles of the selected DEGs, *KCS* (**d**), *DREB1L* (**e**), *CDC* (**f**), *LCHb* (**g**), *AAO* (**h**), *SWEET* (**i**), *HOMEZ* (**j**), *Pcyt-like* (**k**), determined by qRT-PCR.

*3.6. Change in the Expression of Transcription Factors (TFs) Associated with Drought-Stress Response in Common Buckwheat Seedlings*

Transcription factors (TFs) are important for regulating plant response to abiotic and biotic stresses. In the roots of the drought treated seedlings, large numbers of TFs were identified as DEGs, compared to the cotyledons (Figure 6a), and there were 180 TFs that were commonly identified in response to drought stress in both cotyledons and roots. Among them, the most differentially expressed TF families were the C2C2 family, follow by MYB, bZIP, HB and AP2/ERF (Figure 6b, Table S8). According to the GO enrichment analysis, 30.0%, 12.2%, and 9.4% of TFs were classified into "biological regulation", "intracellular", and "nucleic acid binding", respectively (Table S8). In addition, in order to reflect the major trends and patterns, 180 TFs were assigned to six clusters on the basis of their expression patterns. Those in cluster 1 and 2 were up-regulated by drought conditions, but the expression levels of the cluster 1 genes were high at cotyledons, while the cluster 2 genes were highly expressed at roots. Meanwhile, the cluster 3 and 4 genes showed the lowest expression level at cotyledons and roots. The cluster 5 genes were down-regulated by the drought condition. In contrast, there were 25 TFs in cluster 6, and these genes were up-regulated by water deficit, and the genes' expression was most high at cotyledons and roots. These results indicate that the expression of TFs was greatly affected by water deficit in the cotyledons and roots, and had different patterns between cotyledon and root tissues.

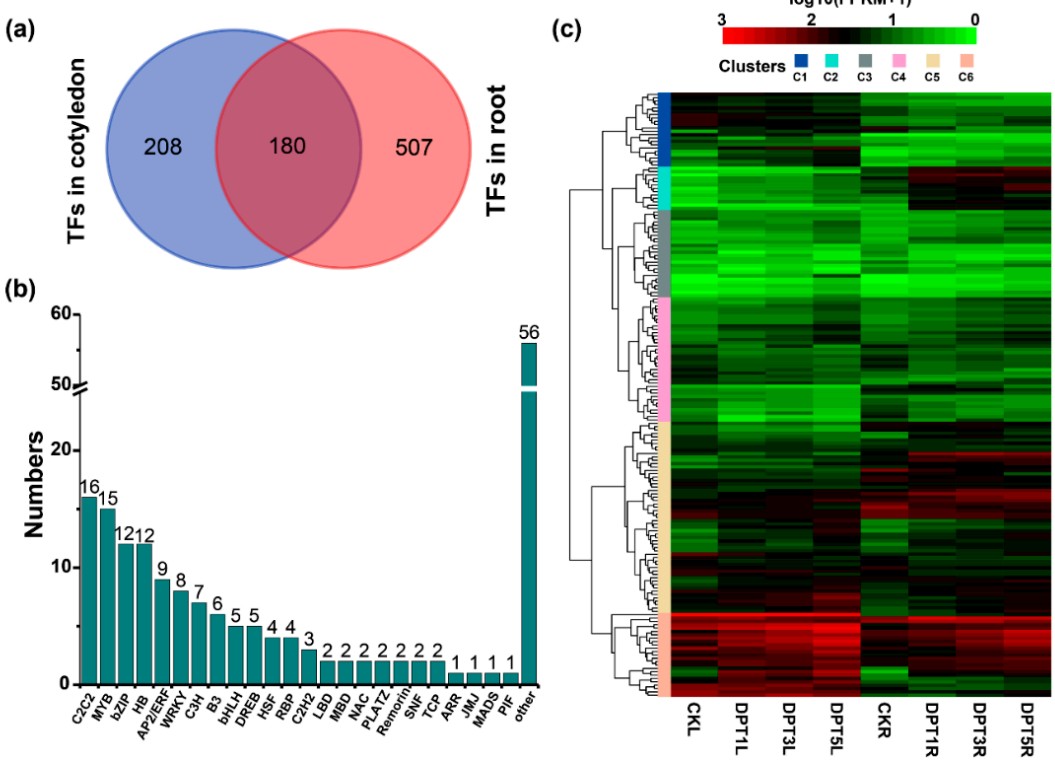

**Figure 6.** The differentially expressed TFs in cotyledons and roots responsive to drought stress. (**a**) Venn diagrams of TFs between cotyledons and roots. (**b**) Classification of TFs that were commonly identified in both cotyledon and root transcriptome libraries. (**c**) Expression pattern of TFs that were commonly identified in both cotyledons and roots in response to drought stress.

## 4. Discussion

Drought stress is one of the most detrimental environmental factors disturbing crop growth and production, and therefore understanding the drought-tolerance mechanism is pivotal for crop breeding [4,43]. Currently, RNA sequencing has been widely used to identify the drought-responsive pathway and genes that are activated during the seedling stage when exposed to abiotic stresses [5].

In this study, the phenotypic and physiological alterations of common buckwheat seedlings during drought stress were analyzed and characterized with the transcriptome analysis. Our results indicate that the common buckwheat seedlings relied on complex biological process to tackle the drought stress.

*4.1. Morphological and Physiological Characteristics Related to Drought Stress in Common Buckwheat*

Under drought stress, plant seedlings exhibit certain physiological and morphological variations [2,8], shown by leaf rolling and wilting [44,45], as well as the decreased RCW and wrinkled leaves (Figure 1a,d). Previous studies have demonstrated that drought stress also inhibits the photosynthesis of plants by affecting chlorophyll biosynthesis and facilitating stomatal closure [46,47], leading to the accumulation of MDA and ROS, which is harmful to the chloroplast photosystem II (PSII) [17,48]. As a result, plants have evolved antioxidant enzyme systems, such as superoxide dismutase (SOD), guaiacol peroxidase (GPX), CAT, and POD, to counteract the damage caused by drought stress [49]. MDA content has been considered important to reflect the drought tolerance ability of plants [12]. An active antioxidant capability in scavenging the cytotoxic ROS is preferred by plants in drought stress [50]. In common buckwheat seedlings, the ratio of chlorophyll a/b and RubisCO activities were significantly decreased compared to the control treatment in 5 days of treatment (Figure 2f, Figure 4e), and the expression levels of DEGs involved in the photosynthesis were correspondingly decreased DPT3L and DPT5L (Figure 4d), which may be due to the decrease in photosynthesis of the common buckwheat seedlings. By contrast, the CAT and POD activities were enhanced (Figure 2b,c). These observations suggest that the drought stress induced evident perturbations in the photosynthesis and ROS scavenging enzyme activities.

*4.2. Multiple Biological Processes Are Involved in Drought Stress Responses in Common Buckwheat*

Previous studies reported that multiple biological processes, such as oxidoreductase activities, and carbohydrate and protein metabolic processes, could be influenced when the ROS accumulation increased [51,52]. Our results indicated that the oxidoreductase activity, kinase activity, and DNA modification were upregulated by the drought treatment in common buckwheat cotyledons, whereas the genes were classified into "light harvesting" and "light reaction" (Figure 4c), and the expression of *LCHb* (encode chlorophyll a/b binding protein) and *PASN* (encode photosystem I reaction center subunit N protein) were markedly downregulated under drought conditions (Figure 4f,l). Repression of photosynthesis under drought stress also occurred in order to help the plants survive the water deficiency [53]. Cellular water deficit in plants caused by drought stress results in weakened carbon fixation, which may be physiologically ascribed to the stomatal closure and the biochemical inhibition of photosynthetic activities, further impacting the carbohydrate metabolism [54]. In addition, phosphorylation, as one of the reversible post-translational protein modification mechanisms, plays an important role in signaling the plant adaptation to osmotic stress [55,56], and a fine-tuned control of protein activity and function [56,57] has also been found to be altered in the common buckwheat root under drought stress. There were 92 DEGs involved in the protein modification and phosphorylation that were up-regulated (Figure 5c, Table S7), which may provide insights in the future study of the protein phosphorylation and modification events in plant drought stress.

*4.3. Genes and Functional Proteins Responsive to Drought Stress*

Previous studies have descried cellular changes that occur upon exposure to drought stress in plants, and the gene responses to drought stress have been studied spaciously in various species [1,9,11,53,58]. Moreover, the plant hormone ABA, as a signal-sensing molecule, can control the expression levels of stress-responsive genes, leading to cellular and physiological changes in response to water deficiency [59–61]. Moreover, previous studies have reported two regulatory systems: ABA-dependent and ABA-independent pathways, that play a major role for plant adaptation to drought stress [62]. Genes upstream and within the ABA pathway can be increased under drought conditions, and the *NCED* gene encoded 9-*cis* epoxycarotenoid dioxygenase has been shown to be induced under dehydration

stress [61]. Furthermore, the changes of several metabolite levels under the water deficit condition were associated with the changes of biosynthetic gene expression, many of which were regulated by the changes of ABA accumulation levels [1,63]. In this study, a large number of DEGs were involved in ABA signaling and regulation (Table S5), and the detection of the up-regulation of the *NCED* genes in this study set identified the candidate genes for further studies of drought resistance in common buckwheat seedlings.

Recently, many drought-inducible genes involved in stress tolerance and stress responses have already been identified in several plant species [1], revealing that the transcription factors (TFs) play a central role in the biotic/abiotic stress responses [64–66]. In our previous study, we isolated and identified a *FeDREB1L* gene encoding a DREB-like transcription factor, which was simultaneously involved in the cold stress, drought stress, and ABA-mediated regulations [31]. The increased expression level of *FeDREB1L* during the earlier stage of drought stress displayed in this study (Figure 4m, Figure 5e) demonstrated that *FeDREB1L* could be a positive factor underpinning the drought stress resistance. Other TFs, including AP2/ERF, MYB, and bZIP families, were also identified to be differentially expressed in this study, for example, 180 DEGs that encoded TFs were identified in response to drought stress in both cotyledons and roots of the common buckwheat seedling, as well as the members of C2H2, MYB, bZIP, and WRKY families (Figure 6a, Table S8). Further studies are thereby required to elucidate the functions and gene-regulatory mechanisms of these TFs in response to plant drought stress.

## 5. Conclusions

To summarize, a comprehensive transcriptome profile of common buckwheat seedlings under drought stress was obtained using RNA-Seq technology. Phenotypic and physiological changes were determined, and the differentially expressed genes were analyzed to understand the regulatory mechanism of common buckwheat seedlings in response drought stress. The photosynthesis of the common buckwheat seedlings decreased, and the activities of antioxidant enzymes such as CAT and POD were increased under drought conditions. DEGs derived from important regulatory metabolisms were characterized. The results reflected in this study may provide useful information to better understand the molecular mechanism underlying the drought resistance in common buckwheat.

**Supplementary Materials:** The following are available online at http://www.mdpi.com/2073-4395/9/10/569/s1, Table S1: List of RT-qPCR primers; Table S2: Phenotypic changes of plant height, root length, and RWC under control conditions across four time points (0, 1, 3, and 5 days); Table S3: Physiological investigation of MDA content, POD and CAT activity, and chlorophyll content under control conditions across four time points (0, 1, 3, and 5 days); Table S4: Summary of the sequencing data of common buckwheat transcriptome; Table S5: Genes related to ABA metabolism in response to drought stress in common buckwheat cotyledons; Table S6: GO enrichment of differentially expressed genes in root transcriptome of common buckwheat seedlings under drought stress; Table S7: Genes related to protein modification and phosphorylation in response to drought stress in common buckwheat roots; Table S8: Classification of TFs that were commonly identified in both cotyledon and root transcriptome libraries; Figure S1: Confirmation of transcriptome data in cotyledons by qPCR analysis; Figure S2: Confirmation of transcriptome data in roots by qPCR analysis.

**Author Contributions:** Z.F., Z.H., and J.Y. designed the study and wrote the manuscript. Z.F., Y.L., J.S., S.W. (Shuping Wang), Z.L., S.W. (Shudong Wei), Y.Z., Z.H., and J.Y. participated in experiments. Z.H. submitted the raw data to Sequence Read Archive (SRA). Z.F., J.S., Z.L., Z.H., and J.Y. discussed the results and revised the manuscript. All authors have read and approved the final manuscript.

**Funding:** This research was funded by the National Natural Science Foundation of China (grant No. 31671755 and No.31571736) and the Supported Project of Outstanding Doctoral and Master's Degree Dissertation Cultivation Program of Yangtze University (YS2018032).

**Acknowledgments:** Authors acknowledge Xiaoyu Xu for critical reading of the manuscript.

**Conflicts of Interest:** The authors declare no conflict of interest.

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
