# Peer review of "Transcriptomic Analysis Reveals the Temporal and Spatial Changes in Physiological Process and Gene Expression in Common Buckwheat (Fagopyrum esculentum Moench) Grown under Drought Stress"

_agronomy, doi:10.3390/agronomy9100569_

Round 1
Reviewer 1 Report
The manuscript written by Hou et al addresses an important topic in common buckwheat regarding the drought stress. The results from the study showed that the photosynthesis decreased in seedlings of the common buckwheat, while ABA pathway might be related to the leaf wilting and thus provided a defense against the water-deficit stress in common buckwheat through a transcriptomic analysis. The information obtained from their study would be particular useful in improving the drought tolerance in common buckwheat.
Some specifics regarding the manuscript:
Line 91-94: The germination method used in the study is not clear. Whether the germination was done in the petri-dishes or in the pots? How long did it take for the roots to reach 2 cm in length under the conditions specified? It would be easier for the readers to have all the information presented.
Line 123-129: More details are needed in the filtering and processing of the sequencing data, such as the cut-off values for quality filtering and the reference for the reference genome.
Line 295: It would be easier to understand why the authors chose this set of genes for qRT-PCR verification of the drought responses in roots if a brief introduction of the genes were included here instead of just “some genes”.
Some other changes were provided in the attached document “summary of comments”.

Author Response
Response to Reviewer 1 Comments
The manuscript written by Hou et al addresses an important topic in common buckwheat regarding the drought stress. The results from the study showed that the photosynthesis decreased in seedlings of the common buckwheat, while ABA pathway might be related to the leaf wilting and thus provided a defense against the water-deficit stress in common buckwheat through a transcriptomic analysis. The information obtained from their study would be particular useful in improving the drought tolerance in common buckwheat.
Some specifics regarding the manuscript:
Line 91-94: The germination method used in the study is not clear. Whether the germination was done in the petri-dishes or in the pots? How long did it take for the roots to reach 2 cm in length under the conditions specified? It would be easier for the readers to have all the information presented.
Response: Thank you very much for your valuable comment. We have added the description “Common buckwheats (cv. Xi’nong 9976) were germinated in the petri-dishes in an incubator (Plant growth incubator JY412L, Shanghai, China) with darkness (25 °C) and relative humidity of approximately 60%. After germination for 36 hours, when the root length of the seedlings was grown approximately 2 cm” in line 89-92 of methods section.
Line 123-129: More details are needed in the filtering and processing of the sequencing data, such as the cut-off values for quality filtering and the reference for the reference genome.
Response: Thank you very much for your kind reminding. We have added the details of the filtering and processing of the sequencing data with the description “The raw reads in FASTQ format were processed using in-house Perl scripts, and the high-quality clean data were obtained by removing the low-quality data, which included the reads that containing adapter, and more than 10% of N nucleotides, and the low-quality reads that containing more than 50% of low quality bases (Q-value ≤ 20). In addition, we calculated the Q30, GC content, and sequence duplication levels for the clean data.” in line 120-124 of methods section.
Line 295: It would be easier to understand why the authors chose this set of genes for qRT-PCR verification of the drought responses in roots if a brief introduction of the genes were included here instead of just “some genes”.
Response: Thank you very much for your kind reminding. We have added the description “The expression of AAO, a gene encode L-ascorbate oxidase which plays a crucial role in plant cell growth, was markedly decreased under drought stress conditions (Figure 5h), and the expression of some transcription factors, like HOMEZ and DREB1L, were significantly induced by water deficit (Figure 5e, j). In addition, the R2 between the two experiments was 0.9385 (Figure S2)” in line 292-296 of results section.
Some other changes were provided in the attached document “summary of comments”.
Response: Thank you very much for your valuable comment. We have revised the article according to the attached document “summary of comments”, and the modifications were highlighted.

Reviewer 2 Report
The manuscript by Hou et al. describes transcriptome analysis of stress phenotypes of buckwheat seedlings under drought condition, to identify differentially expressed genes (DEGs) in response to drought. In general, the manuscript is well-written, however, it is somewhat like a descriptive report describing what they observed through analyzing the data. Further analyses should be done to explain what they have observed and to reveal “the temporal and spatial changes in physiological process” using transcriptome data as the title reads.
Please find my comments below:
The supplementary materials are not available with this version. Line 91, line 103, 108 and many other sentences throughout the manuscript: it is probably better not to start a sentence with the word “And”. The sequencing read data is not yet available using the BioProject accession number (PRJNA555746) provided. Line 119: it is not consistent when the authors referred to quality score here. These included Phred score, Q30, and then Q-value. Are these referring to the same score? Line 125: what is the definition of “resulting sequences” here? Were these Coding genes (CDS) annotated on the reference genome that the authors mentioned in line 194? I think Figures 1 and 2 would be more convincing if the authors provide data of control samples after 1d, 3d and 5d, together with the treated samples. Since the treatment period was 5d long, what the authors think about the differences in MDA, POD, CAT and chlorophyll in control samples, after 1d, 3d, and 5d in the control non-treated samples? Would any of the mentioned change overtime, e.g., 3d-, 5d- control seedlings? Line 191: “88.76 Gb of clean bases”, is this Gigabytes or Gigabases? Change to "clean reads"? Please indicate read length. For DEG analysis, according to the methods, the authors mapped the RNA-Seq reads against the reference genome to obtain FPKM values, I do not really understand what the meaning of 877,111 CDS (line 194), how were these obtained? Were these reference transcriptome used in transcript profiling? The length distribution seems to be short to me. Line 192: what was the BLAST setting? I think it would be more meaningful if the authors provide % of read from each sample mapped to the reference genome, rather than BLAST results. Figure 3 legend: please explain in the figure legend what are CKL, DPT1L and so.! Similar to comment 6: in Figure 3, the authors compared the treated samples (1d, 3d and 5d) against control sample (non treated, 0d) and identified DEGs. Would the DEGs associated with development and independent from drought responses also be detected? The use of control sample +1d, +3d and +5d collected at the same developmental stages with the treated samples would exclude those DEGs related to development. I think the stress related DEGs were present in the total DEGs, but so were those developmental DEGs. Did the authors consider this possibility? This might affect the result interpretation including the 180 shared transcription factors. Line 274: “screened DEGs”, do you mean “identified DEGs”? or non-overlapped DEGs? Line 282: what does “Mapman analysis were up-regulated after drought treatment” mean? Figure 5: The authors should indicate in the plots whether the expression levels were significantly different. Also, for qPCR in generally in this work, it is difficult to judge whether they were consistent with the corresponding RNA-Seq data, unless the RNA-Seq data was also included in the plots. The authors could include RNA-Seq FPKM values for each plot, and or calculate the correlation coefficient between the two methods to see whether they are consistent. Line 306: It is not obvious in figure 6 that “the expression of TFs was greatly affected by water deficit in the cotyledons and roots”, because I cannot see the trend by just looking at the heatmaps. The authors might consider plotting the data using row-scale values to highlight the trends (e.g., highlight the up-regulation in the treated samples). For the 180 TFs, I think the authors should explore their potential functions, and identify those TFs that were known to be involved in drought responding. What about those tissue-specific TFs, that were only detected in root or cotyledon? What are the potential functions of those TFs? The analysis for this work would not be complete and thorough if the authors just stopped here. Without further analyses, the manuscript is just a descriptive report describing what the authors observed through some tests. Further analyses could be done to explain what the authors observed.Author Response
Response to Reviewer 2 Comments
The manuscript by Hou et al. describes transcriptome analysis of stress phenotypes of buckwheat seedlings under drought condition, to identify differentially expressed genes (DEGs) in response to drought. In general, the manuscript is well-written, however, it is somewhat like a descriptive report describing what they observed through analyzing the data. Further analyses should be done to explain what they have observed and to reveal “the temporal and spatial changes in physiological process” using transcriptome data as the title reads.
Please find my comments below:
The supplementary materials are not available with this version.
Response: Thank you very much for your kind reminding. There was a mistake when we uploaded the supplementary materials, and the supplementary materials are available with this version.
Line 91, line 103, 108 and many other sentences throughout the manuscript: it is probably better not to start a sentence with the word “And”.
Response: Thank you very much for your kind reminding. We have removed the word “And” and use other connectives to start a sentence.
The sequencing read data is not yet available using the BioProject accession number (PRJNA555746) provided.
Response: Thank you very much for your valuable comment. When we submit the sequencing data, we made a setting, and the sequencing read data will available at NCBI since 1st October, 2019.
Line 119: it is not consistent when the authors referred to quality score here. These included Phred score, Q30, and then Q-value. Are these referring to the same score?
Response: Thank you very much for your kind reminding. We have modified these sentences with the description “The raw reads in FASTQ format were processed using in-house Perl scripts, and the high-quality clean data were obtained by removing the low-quality data, which included the reads that containing adapter, and more than 10% of N nucleotides, and the low-quality reads that containing more than 50% of low quality bases (Q-value ≤ 20). In addition, we calculated the Q30, GC content, and sequence duplication levels for the clean data.” in line 120-124 of methods section.
Line 125: what is the definition of “resulting sequences” here? Were these Coding genes (CDS) annotated on the reference genome that the authors mentioned in line 194?
Response: Thank you very much for your valuable comment. The “resulting sequences” in line 125 refers the sequences that were subjected to functional annotation and coding sequence (CDS) prediction, and the CDS that mentioned in line 194 were annotated on the reference genome. We have added the description “Then these sequences were subjected to functional annotation and coding sequence (CDS) prediction [41], and the resulting sequences were called genes.” in line 125-126 of methods section
I think Figures 1 and 2 would be more convincing if the authors provide data of control samples after 1d, 3d and 5d, together with the treated samples. Since the treatment period was 5d long, what the authors think about the differences in MDA, POD, CAT and chlorophyll in control samples, after 1d, 3d, and 5d in the control non-treated samples? Would any of the mentioned change overtime, e.g., 3d-, 5d- control seedlings?
Response: Thank you very much for your valuable comment. We add the data of control samples after 1d, 3d and 5d, together with the treated samples in Table S1 and Table S2 The description as “there was no significantly change at plant height, root length and RWC of control samples after 1d, 3d, and 5d (Table S2)” was added in line 158-159, and description as “Compared with the drought treatment, these physiological traits were not significantly changed during the whole treatment under control condition, which the content of chlorophyll a and chlorophyll b were increased in the 1d-, 3d- and 5d-control plants, but the chlorophyll a/b ratios was no significantly changed during the whole treatment under control condition (Table S3).” was added in line 175-178.
Line 191: “88.76 Gb of clean bases”, is this Gigabytes or Gigabases? Change to "clean reads"? Please indicate read length. For DEG analysis, according to the methods, the authors mapped the RNA-Seq reads against the reference genome to obtain FPKM values, I do not really understand what the meaning of 877,111 CDS (line 194), how were these obtained? Were these reference transcriptome used in transcript profiling? The length distribution seems to be short to me.
Response: Thank you very much for your kind reminding. We have modified these sentences with the description as “88.76 Gigabytes (Gb) of clean reads were abstained” in line 198 of results section, and “Then these sequences were subjected to functional annotation and coding sequence (CDS) prediction [41], and the resulting sequences were called genes. Finally, fragments per kilobase of transcript permillionmapped reads (FPKM) method was used to calculate the gene expression unit” in line 125-126 of method section.
Line 192: what was the BLAST setting? I think it would be more meaningful if the authors provide % of read from each sample mapped to the reference genome, rather than BLAST results. Figure 3 legend: please explain in the figure legend what are CKL, DPT1L and so.!
Response: Thank you very much for your valuable comment. We changed “blast efficiency” to “mapped efficiency”, and added the legend of CKL, DPT1L, DPT3L, DPT5L, CKR, DPT1R, DPT3R and DPT5R in line 226-230.
Similar to comment 6: in Figure 3, the authors compared the treated samples (1d, 3d and 5d) against control sample (non treated, 0d) and identified DEGs. Would the DEGs associated with development and independent from drought responses also be detected? The use of control sample +1d, +3d and +5d collected at the same developmental stages with the treated samples would exclude those DEGs related to development. I think the stress related DEGs were present in the total DEGs, but so were those developmental DEGs. Did the authors consider this possibility? This might affect the result interpretation including the 180 shared transcription factors.
Response: Thank you very much for your kind reminding. We also consider this possibility, and previous studies had indicated that stress will delayed plant development. According to the phenotypic and physiological changes at this study, we finally chose non-treated seedlings as the control sample. In addition, we also carried out the GO and KEGG enrichment analysis to exclude the DEGs that associated with development, and focus on the stress-response DEGs.
Line 274: “screened DEGs”, do you mean “identified DEGs”? or non-overlapped DEGs?
Response: Thank you very much for your kind reminding, we changed “screened DEGs” to “non-overlapped DEGs” in line 281 of results section.
Line 282: what does “Mapman analysis were up-regulated after drought treatment” mean?
Response: Thank you very much for your valuable comment. we have modified the sentence with the description as “According to Mapman software analysis, there were 92 protein modification and phosphorylation related DEGs were up-regulated after drought treatment (Table S7) suggesting that they are candidate genes for protein modification in roots of drought-treated seedlings” in line 287-290.
Figure 5: The authors should indicate in the plots whether the expression levels were significantly different. Also, for qPCR in generally in this work, it is difficult to judge whether they were consistent with the corresponding RNA-Seq data, unless the RNA-Seq data was also included in the plots. The authors could include RNA-Seq FPKM values for each plot, and or calculate the correlation coefficient between the two methods to see whether they are consistent.
Response: Thank you very much for your kind reminding. The correlation coefficient between RNA-Seq data and qPCR results was calculate, and submitted as Supplementary Figure S1 and Supplementary Figure S2. The description as “The correlation coefficient (R2) between RNA-Seq data and qPCR results for 24 total plots was 0.8743 (Figure S1)” was added in line 262-263, and description as “The R2 between the two experiments was 0.9385 (Figure S2)..” was added in line 295-296.
Line 306: It is not obvious in figure 6 that “the expression of TFs was greatly affected by water deficit in the cotyledons and roots”, because I cannot see the trend by just looking at the heatmaps. The authors might consider plotting the data using row-scale values to highlight the trends (e.g., highlight the up-regulation in the treated samples). For the 180 TFs, I think the authors should explore their potential functions, and identify those TFs that were known to be involved in drought responding. What about those tissue-specific TFs, that were only detected in root or cotyledon? What are the potential functions of those TFs? The analysis for this work would not be complete and thorough if the authors just stopped here. Without further analyses, the manuscript is just a descriptive report describing what the authors observed through some tests. Further analyses could be done to explain what the authors observed.
Response: Thank you very much for your kind reminding. Further analyses was carry out. The details information of these 180 TFs was show in the Table S8, and we modified the heatmaps in Figure 6 to make it easy to see the trend change that effect by drought condition. The description as “According to the GO enrichment analysis, 30.0% (54), 12.2% (22) and 9.4% (17) TFs were classified into “biological regulation”, “intracellular” and “nucleic acid binding”, respectively (Table S8). In addition, in order to reflect the major trends and patterns 180 TFs were assigned to 6 clusters based on their expression patterns. The cluster 1 and 2 up-regulated by drought conditions, but the expression levels of the cluster 1 genes were highly at cotyledons, while the cluster 2 gene were highly expressed at roots. Meanwhile, the cluster 3 and 4 genes shown lowest expression level at cotyledons and roots. The cluster 5 genes were down-regulated by drought condition. In contrast, there were 25 TFs in cluster 6, and these genes were up-regulated by water deficit and the genes expression were most highly at cotyledons and roots. These results indicate that the expression of TFs was greatly affected by water deficit in the cotyledons and roots, and had different patterns between cotyledon and root tissues.” was added in line 315-325.
Round 2
Reviewer 2 Report
The authors have addressed all of my concerns appropriately.
The manuscript has been improved.